# Peer Intervention following Suicide-Related Emergency Department Presentation: Evaluation of the PAUSE Pilot Program

**DOI:** 10.3390/ijerph20043763

**Published:** 2023-02-20

**Authors:** Mandy Gibson, Nick Moreau, Eschleigh Balzamo, David Crompton

**Affiliations:** 1Australian Institute for Suicide Research and Prevention, WHO Collaborating Centre for Research and Training in Suicide Prevention, School of Applied Psychology, Griffith University, Brisbane, QLD 4122, Australia; 2Brook Red Mental Health Charity Ltd., Brisbane, QLD 4122, Australia

**Keywords:** peer work, peer specialist, suicide prevention, post-hospitalisation, assertive outreach

## Abstract

The risk for future suicidal behaviours is elevated following suicide attempts, particularly for those with complex needs or those who are disconnected from healthcare systems. The PAUSE program was designed to address this gap using peer workers to provide continuity and coordination of care following suicide-related emergency presentations. This study aimed to evaluate the pilot program’s effect on suicidal ideation and hope, and to explore the acceptability and participants’ experiences. A mixed-methods design was employed with pre- and post-evaluation questionnaires, including the GHQ-28-SS (general health questionnaire suicide scale), AHS (adult hope scale), and K10 (Kessler psychological distress scale). Participant engagement rates and semi-structured interviews were used to explore program acceptability. In total, 142 people were engaged with the PAUSE pilot between 24 August 2017 and 11 January 2020. There were no significant gender differences in engagement. The suicidal ideation scores decreased, and the hope scores increased after participation in PAUSE. A thematic analysis revealed that participants identified that the key program mechanisms were holistic and responsive support, ongoing social connectedness, and having peer workers who understood their experiences and treated them like people rather than clients. The small number of participants and lack of a control group limited the result generalizability. The findings suggest that PAUSE was an effective and acceptable model for supporting people following suicide-related hospitalisations in this pilot sample.

## 1. Introduction

A previous suicide attempt remains the strongest predictor of future death by suicide [1]. Over a third (35.2%) of those who died by suicide in Queensland between 2016 and 2018 had a prior suicide attempt during their lifetimes and 17% had a non-fatal attempt in the previous year [2]. The risk of repeated suicidal behaviours increases in the period following the attempt and hospital discharge, especially for people with complex needs or those who are not well connected to community or healthcare systems [3]. Though repeated behaviours occur predominantly in the following weeks, this risk can remain elevated for several years [4,5,6]. Supporting people after an attempt is a critical component of suicide prevention efforts [7]. Assertive follow-up care and support are promising models for reducing future suicidal behaviours [8,9], as they provide a window of opportunity to connect people with necessary services. The emergency department can be the first point of contact for intervention for many people, especially for people who have experienced suicide attempts in the context of environmental or situational stressors without symptoms of mental health disorders [10]. 

An issue identified by people who have experienced suicidality is that the treatment they received often focused on mental illness diagnoses or risk assessments to the exclusion of precipitants or triggers contributing to their suicidality or the contextual stressors they experienced [11,12]—despite the well-documented association between social and environmental contextual factors, such as poverty, unemployment, interpersonal violence, and increased suicide risk [13,14]. Indeed, previous research has reported that people experiencing these adverse life events and stressors are more likely to die in the weeks following discharge from psychiatric admissions than those who are discharged who were not also experiencing such stressors [15]. Though suicidality and mental health issues often co-occur, suicide is a behavioural phenomenon rather than a mental illness disorder, the aetiology of which is as varied and complex as each individual person impacted [10,16,17]. As such, post-hospitalisation and post-attempt interventions need to be able to respond to this complexity and broad range of contexts and precipitants. 

### 1.1. Peer Worker Intervention Models

Peer workers—people who have their own lived experiences of mental ill-health and are trained to provide support to others—have been included within mental health services for decades [18,19]. During this time, findings have broadly found peer-support services to be as effective at symptom reduction as traditional clinician-delivered services [20], with some reporting peer-specialist programs to be more effective at strength-focused and recovery-oriented outcomes such as empowerment, hope, and quality of life [21,22]. Peer support has been proposed as a promising model for supporting people experiencing suicidal crises and providing person-centred post-suicide attempt and post-hospitalisation care [23], as core peer work actions such as teaching coping and problem-solving skills, addressing social isolation, connecting people to community resources, advocating and liaising between services, and recovery planning have been identified as essential by people experiencing suicidal crises [24,25,26,27]. In addition to sharing symptom-management strategies, people with their own lived experience are often uniquely placed to provide guidance on dealing with stigma and navigating often complex or disjointed community and health systems [28,29,30]. As those who have a history of suicide attempts face unique experiences of stigma [27,31], peer workers may be uniquely effective at reducing the associated barriers and harms of social and self-stigmatisation. The evidence for the effect of peer worker interventions on key risk factors for suicidality, such as isolation, is promising [32,33,34]. 

Although few program evaluations have examined the effect of peer-support interventions on consumers’ suicidality, peer workers are increasingly utilised within suicide prevention settings, such as crisis lines and hospital services [9,28,35,36]. Peer worker inclusion within mental health services has indicated positive outcomes, including decreased emergency service use for individuals with a history of multiple attempts, participant satisfaction, patient acceptability, and cost efficiencies [25,35]. Few studies, however, have explored the role of peer workers in suicide prevention to understand the underlying mechanisms for why these models may reduce suicidality [37].

### 1.2. Brook RED 

Brook RED is a mental health charity that has operated in Southeast Queensland since 2000. Brook RED is a wholly lived experience-governed and -operated service provider; every person who works at Brook RED has had a personal experience of their own mental ill-health, suicidality, and/or the mental health system [38]. The unique perspectives and knowledge gained from these experiences are foundational to all of Brook RED’s activities as a growing community of lived experiences. The lived experiences of peers are conceptualised as a mechanism to challenge stigma and hopelessness through normalising the common experiences of mental illness, with these shared experiences facilitating honest and effective discussions [38,39,40]. Peers are seen as providing ongoing credibility to confirm to people that it is possible to live through mental illness [23,25]. Brook RED delivers peer-support connection programs and services, disability support, anti-stigma community forums, and residential spaces where people can spend several weeks working toward recovery goals, amongst other services supporting people experiencing mental ill-health.

### 1.3. PAUSE Program

The PAUSE pilot program was developed and implemented by Brook RED to support people in the critical period following emergency presentations with suicidal ideation, a suicide attempt, or an episode of self-injury [38,41]. The pilot project sought to use the lived experience of peer workers to provide holistic and responsive support and short-term continuity and coordination of care (up to 13 weeks) to people presenting to the Logan General Hospital—Emergency Department. 

PAUSE provided a single point of contact to hospital staff for post-presentation referrals, after which PAUSE workers would attempt to contact the referred individual by phone within 24 to 72 h, as seen in Figure 1. PAUSE workers provided referred individuals with the opportunity to talk about their suicidal experiences with workers, and also shared recovery strategies that they had utilised themselves and appropriately shared their own stories. The peer workers also assisted participants with identifying and making goals toward change, and finding resources and people who would support them in this process. This often involved non-clinical or unfunded supports in the community. Peer workers also provided practical support such as transportation, liaison, advocacy, and any other support and assistance required during this time and assisted people with engaging with the required health and community services [38]. 

The current study aimed to: (1) evaluate the effectiveness and acceptability of the PAUSE pilot program of using peer specialists to reduce suicidality and mental illness symptoms, and increase hope for people following a suicide-related hospital presentation, and (2) explore the experiences of people receiving peer-support follow-up after an emergency department presentation for self-harm, suicidal attempts, or suicidal thoughts to understand key factors contributing to the impact or outcomes of the program.

## 2. Materials and Methods

The evaluation methodology was co-designed with peer workers using their own lived experiences of mental ill-heath and/or suicidality to ensure a consumer-oriented approach across all methodologies. A mixed-methods approach was used to cover the domains of program impact and acceptability.

### 2.1. Participants and Study Population

All people who presented to the Logan General Hospital—Emergency Department for suicidal ideation, self-injury, or a suicide attempt were provided with information about the PAUSE program and were invited to engage with a peer worker to provide follow-up after presentation. Those who consented were referred to PAUSE by emergency department staff, who provided the participant contact details to the PAUSE staff. The PAUSE staff contacted all participants who were referred within 24 to 72 h to arrange a face-to-face contact within a week post-presentation.

The Logan General Hospital is the primary hospital serving Logan City, one of the most diverse and fastest-growing cities in Australia and home to approximately 350,000 people in Southeast Queensland, Australia [42]. Compared to the national and state-level averages, people residing in Logan experience disadvantages across multiple domains, including income, health, and educational outcomes [42].

### 2.2. Evaluation Questionnaire

Demographic information, including gender, age, Indigenous status, CALD (cultural and linguistic diversity) identification, and referral reason, was collected for all participants who were referred to PAUSE for comparison in the populations who engaged—chose to participate—with a PAUSE worker. At the first face-to-face contact, all participants were invited to participate in the evaluation study and complete the pre-program evaluation questionnaire. The participants were asked to complete the post-program evaluation questionnaire between 9 and 13 weeks after first engaging with a PAUSE worker, unless the participant exited the service prior to this point. The evaluation questionnaire included the following measures:

#### 2.2.1. General Health Questionnaire Suicide Scale (GHQ-28-SS)

Suicidal ideation was measured using the 4-item GHQ-28-SS, which assesses ideation over the previous 4 weeks and has been previously used in Australian samples [43]. This study followed the design used by Goldney et al. [44], in which a response of 2 or 3 on any item is recorded as current ideation. 

#### 2.2.2. Adult Hope Scale (AHS)

As strength-focused orientation is a key principle of peer support [22], the study design included positive or recovery measures. The adult hope scale is a widely used positive psychology instrument that has been included within previous peer-support program evaluations [45]. Eight items are rated on an 8-point scale from “definitely false” to “definitely true” and summed to produce a total score. AHS scores have been positively associated with coping strategies, treatment adherence, service attendance, and engagement and negatively associated with depression and suicidal ideation [46]. 

#### 2.2.3. Kessler Psychological Distress Scale—10 (K10)

The K10 is a widely used instrument in both clinical and community samples to measure broad anxiety- and depression-type psychological distress over the previous 4 weeks on a 5-point Likert scale [47,48,49,50]. Scores above 30 reflect a high probability of current severe mental ill-health symptomology [49].

### 2.3. Statistical Analyses

Chi-square statistics were calculated to compare the proportion of referred participants who elected to engage with PAUSE by gender, Indigenous status, CALD identification, and reason for referral. Non-parametric Wilcoxon signed-rank tests were used to compare the changes between pre- and post-intervention, as normality could not be assumed in the pilot sample. Significance levels of 0.05 were used for both analyses.

### 2.4. PAUSE Experience Questionnaire

A questionnaire that included items asking participants about their perceived effectiveness and satisfaction with PAUSE and open-ended items on service improvement was given post-intervention/at the service exit. The questionnaire also included the 9 peer support tasks and technique items that were identified by peer workers from Brook RED as unique, specific actions differentiating peer support from alternative models of support or follow-up (i.e., my peer worker understood my experiences because they have had similar experiences, and my peer support worker discussed strategies they have used) [38]. Participants were asked if their peer worker engaged in each of the identified actions and how helpful participants found the actions to be on a 4-point Likert scale from “not at all” to “very helpful.” These items share considerable overlap with the core peer specialist roles, actions, and processes as reported in the peer fidelity assessments developed by Chinman and colleagues [51]. 

### 2.5. Semi-Structured Interviews 

Semi-structured individual interviews were conducted to explore participants’ experiences, insights, and reflections regarding receiving support from a peer worker with their own experiences of mental illness and suicidality, the assistance provided by PAUSE workers, the most and least beneficial aspects of the PAUSE model, the barriers, the limitations, and possible program improvements. While the interviews were semi-structured in nature with scope to respond to the participants’ direction, the key interview questions included: What support or assistance did your PAUSE worker provide? What was your experience with being supported by a peer worker with their own lived experience? What could be implemented to improve the program going forward? Would you want someone you cared about to access PAUSE if they experienced a suicidality crisis? Though the interview questions did not focus on events leading up to the initial presentation, all interviews were conducted by a researcher and psychologist possessing clinical expertise with people experiencing suicidal crises. Participants were provided with safety procedures and crisis support options and were able to be followed up on by their original peer worker. Participants were offered to have the interview conducted over the phone, in-person at a location of their choosing, or at the researcher’s offices. One participant chose to be interviewed at the researcher’s offices with the remaining all electing to be interviewed over the phone. The interviews were recorded and transcribed verbatim. A thematic analysis was utilised to identify key themes emerging from within the transcribed data. This was performed by broadly following Braun and Clarke’s guidelines [52]. 

## 3. Results

### 3.1. Participants

Between August 2017 and January 2020, 315 people were referred from the emergency department of the Logan General Hospital following suicide-related presentations. One-hundred and forty-two (142) people (45.1%) of those referred engaged with a PAUSE worker; 88 (62%) were women and 54 (38%) were men; and the mean age of the people who engaged was 32.69 (*SD* = 13.99). A total of 42.5% of the men and 47.3% of the women referred engaged with a PAUSE worker. A chi-square test found no significant difference in engagement by gender (X^2^(1) = 1.528, *p* = 0.466). As seen in Table 1, a quarter (25%) of First Nations people who were referred engaged with PAUSE, compared to 48.7% of non-Indigenous people. This difference approached significance (X^2^(1) = 3.376, *p* = 0.066). 

A total of 59.5% of people from culturally and linguistically diverse populations engaged with a PAUSE worker, compared to 45.3% of people from other backgrounds, though this was not significantly different (X^2^(1) = 1.615, *p* = 0.204). A total of 39.7% of people referred with a history of mental illness engaged with a peer worker, compared to 50.6% of those presenting to the emergency department experiencing a situational crisis. A total of 50% of people who presented in the context of both situational precipitants and a mental illness history engaged with PAUSE. The likelihood of engagement by referral background/reason was not statistically different (X^2^(2) = 1.887, *p* = 0.389).

### 3.2. Evaluation Questionnaires

Evaluation questionnaires were collected for 54 participants (38.3% of all participants who engaged and 55.1% of people who engaged in in-person contact); 33 “pairs” of pre- and post-program evaluation questionnaires were able to be “matched” for analysis. Fewer (*N* = 21) were available for the K10, as several people chose not to complete this scale. Several participants reported frustration or distress at being asked to complete the K10, with comments including “*I just did this at the hospital*” or “*we are always filling this one out.*”

#### 3.2.1. Suicidal Ideation 

As seen in Table 2, the GHQ-28-SS mean score from the 53 questionnaires that were received pre-intervention was 9.15 (*SD* = 3.2). Almost all (92.5%) participants who completed the questionnaire pre-intervention reported current suicidal ideation. As seen in Table 3, the GHQ-28-SS mean score decreased significantly from 9.30 to 4.52 post-intervention (*Z* = 4.842, *p* < 0.001). An exact McNemar’s test found that the proportion of participants experiencing current suicidal ideation also fell significantly after engaging with a PAUSE worker (*p* = 0.001).

#### 3.2.2. Hope

The mean pre-intervention/baseline AHS score was 31.91 (SD = 14.16) for all 53 pre-intervention questionnaires collected. The mean score increased from 32.49 to 39.19 after participating in the PAUSE program (*Z* = 2.789, *p* = 0.005) for the 32 people for whom pre- and post-program questionnaires were collected.

#### 3.2.3. Psychological Distress

The mean pre-intervention/baseline K10 score exceeded the threshold to indicate probable severe mental health symptomology for the 36 people for whom pre-program questionnaires were collected, with 37.58 (SD = 8.09); 86.1% of the individuals who completed a baseline survey scored above this level. The mean psychological distress score decreased from 38.48 to 32.05 after the program, which was significant in a sign test (*Z* = 2.627, *p* = 0.009) for the 21 people for whom pre- and post-intervention questionnaires were received. However, an exact McNemar’s test determined that the decrease in the proportion of people with psychological distress scores above levels indicating probable current severe mental health disorders was not statistically significantly different (*p* = 0.125).

### 3.3. PAUSE Experience Questionnaire 

PAUSE experience questionnaires were collected from a total of 36 PAUSE participants. All the respondents reported that they believed the program had been beneficial to them and would want someone they cared about to receive support from the PAUSE program if they were experiencing a suicidal crisis. As seen in Table 4, all participants who completed this questionnaire reported that they felt a sense of connection with their PAUSE support worker and that their PAUSE worker made them feel that their recovery work was valuable and valid. For each of the nine key peer skills and experiences, at least 82% of the respondents identified that their peer worker engaged in these activities. More than one in six participants reported that their PAUSE worker did not discuss their challenges with mental illness or their recovery.

Of the participants who reported that their peer worker did engage in these actions, 82.8% reported that their peer worker made them feel that their work towards recovery was valid and valued it as “very helpful.” A sense of belonging with their peer worker and a sense of connection with their peer worker was reported as “very helpful” to 77.8% and 72.4% of participants, respectively. However, 10.3% and 7.4% of participants reported that knowing their peer worker had overcome challenges and the peer worker’s sharing strategies they had tried or used were “not at all” helpful, respectively.

### 3.4. Semi-Structured Interviews 

Ten previous PAUSE participants (six men and four women) were interviewed. The interviewees were selected to reflect a balance of gender, age, type of PAUSE support provided, and mental health diagnoses prior to the suicide-related incident. All ten interviewed participants reported that PAUSE had benefitted them, with a few reporting the program as a significant contributor to avoiding recurrent hospitalisations or even to keeping them alive.


*“I got referred to the PAUSE program and I’ve now actually reached that point where I’ve not hurt myself at all.”*


A number of interviewees reflected that they probably would have avoided needing emergency presentations or hospitalisation had they received this type of assistance or support in their life: “*Maybe if I’d been connected with them earlier then maybe something would have happened differently.*”

The thematic analysis revealed four key themes as critical factors underlying the PAUSE program’s effect: holistic and responsive support; peer workers understanding the participants’ experiences; treating the participants like people rather than clients; and ongoing social connectedness.

#### 3.4.1. PAUSE Workers Provided Holistic and Responsive Assistance

Participants described PAUSE workers as providing a wide range of support, including helping with disability assistance applications, advocating and liaising with service providers, helping with applications to the Department of Housing waitlists, booking and transporting participants to mental health and medical appointments, coordinating cancer treatments, reviewing legal statements, assisting with job searches, applying for death certificates for family, communicating with relatives, counselling, and providing other social and emotional support.

As participants framed the stressors contributing to their suicidal crises as complex, multifaceted, and fluctuating, the interviewees in turn reflected that the individual and holistic model whereby PAUSE workers responded to their needs in many areas was critical to the effect on their suicidality.


*“She’s setting up an appointment for me to get some financial counselling. She has set up appointments with my psychologist and my psychiatrist, but like she just wants to help, so she was like, well sounds like you need a medication review with some counsellors. I gave her their information and she helped to arrange everything, that was very helpful.”*


The PAUSE workers were seen as responsive to people’s needs as they changed over time, as opposed to offering predetermined programs or activities, and were able to stay connected from periods of high acuity and lower-intensity symptoms and distress. Participants identified that immediately after leaving the hospital, people required more support and greater involvement and that this generally decreased and changed over time.


*“I haven’t actually reached that point as often where I really, really need to go to them. To the point where I really sort of contact for moral support sort of, “have you been through this, how did you deal with it? Okay, do you reckon that might work for me? You know. Just a casual talk.”*


Several participants recounted their difficulty in coordinating with multiple providers or even finding the services they needed while they were experiencing a suicidal crisis. Peer workers were able to coordinate and liaise with providers on their behalf when it was too triggering, overwhelming, or would risk exacerbating their acute suicidality. 

#### 3.4.2. PAUSE Workers Better Understood Participants Due to Their Lived Experience

Many people identified that their PAUSE worker was able to better understand their experiences because they had their own experiences and history of suicidality and mental illness. This was described as central to participant’s improvement, as they felt it allowed them to trust their peer worker and feel comfortable to share, as “*it’s not like you’re talking to someone who has no clue of what it feels like. It’s a little bit easier usually talking to someone when you know they’ve been through something similar.*”

As PAUSE workers understood participants’ experiences, they, in turn, asked more relevant and salient questions. One person reported that their PAUSE worker “*got me more directly than anyone else.*” Notably, this included discussing and addressing internalised stigmas that had hindered their recovery previously, as they felt able to “*express some of the deeper issues that have me sort of feeling ashamed of my issues.*”

#### 3.4.3. Treated like a Person Rather Than a Client by PAUSE Workers

The interviews identified that PAUSE workers treated the participants like “individuals” or “people” rather than service provider “clients”: “*they’re more focused on the actual helping out rather than just getting a job done.*” The peer workers were described as being genuinely concerned about them as individuals and “*actually interested in getting to know me*,” which many reported they did not experience with other providers where they felt seen as “*just a number*.” 

The participants reported feeling able to share more broadly and talk more informally about their lives, in contrast to only targeted conversations about their illness or symptomology, which they reported in other models: “*I just sort of I bounce ideas off him and he’s genuinely interested in what’s going on in my life.*” Several interviewees identified that they were describing actions that normally would—or should—ideally be performed by family or friends. Several participants reflected that peer support was uniquely effective to their outcomes, as they may not have experienced the suicidal crisis they did if they had people in their lives to perform these roles: “*We ended up where we did because there’s no one, we’ve had no support at all, like no family or friends. Having that one person that does care helps a lot.*”

Some participants clarified further that it was not simply a lack of people in their lives, but that the people in their lives may be unable to provide this support due to their own stressors or mental illness: “*That’s how [PAUSE] seems to work. It’s almost friend-like without actually having that interpersonal relationship involved*.”


*“It’s not like a formal setting where you’re with a psychologist or something, it’s like a weird in-between, just talking to someone, and talking to a professional. It’s a lot calmer and you feel like you can open up a bit better.”*


#### 3.4.4. PAUSE Facilitated Ongoing Social Connection

The interviewees reported that a key way in which PAUSE helped them was by building social connections and addressing isolation, which many described as a contributor to their suicidality: “*I sort of came from not having much support apart from a psychologist, that’s all I really had for many, many years.*”

For some interviewees, PAUSE was described as their primary, or occasionally sole, form of social support or connection: “*I’m pleased [PAUSE worker] comes around because he’s the only one that does*.” 

Though most people were not receiving direct support from a PAUSE worker at the time of the interviews, the positive effect of the program was described by many interviewees in a continual sense; i.e., they felt they were at lower risk of future suicidal behaviours because they were “now” connected to people at Brook RED that they could contact if they needed assistance or support in the future.


*“If I thought I unfortunately may end up back in hospital, I know that there’s always that number. I find that comforting. Whereas with a psychologist and stuff like that, you obviously can’t do that.”*


Several participants reported that they felt part of the Brook Red community now and discussed further involvement with the organisation to assist and “*meet some people that are going through the same stuff.*” Some described supporting other people as peer mentors to support their recovery.

## 4. Discussion

The purpose of this study was to evaluate the effectiveness and acceptability of the PAUSE pilot program using peer workers to provide support to people after suicide-related emergency department presentations and to explore the experiences of people receiving peer-support follow-up in these contexts. This aim was achieved by: examining participant engagement rates of priority populations; using a pre-/post-program evaluation questionnaire to examine changes in suicidal ideation, hope, and psychological distress; and conducting participant interviews and post-program surveys to explore participants’ perceptions of the pilot program, and experiences of peer worker support. The engagement rates were not significantly different for men, people from CALD populations, or people experiencing situational crises, suggesting that PAUSE is an acceptable model for people from priority populations. As all interview and survey participants reported that the program had been beneficial, and their suicidal ideation and hope scores improved significantly after the program, the findings indicate that this peer-support program was an effective model for supporting people during the critical and challenging period following suicide-related emergency hospitalisation. Participants’ suicidal ideation scores decreased significantly after engaging with a PAUSE worker, as did the proportion of participants reporting current suicidal ideation. This decrease occurred in the weeks following a suicide attempt or suicide-related hospitalisation, which is a period associated with an elevated risk of further and even fatal suicidal behaviours [4]. While this result must be understood in the context of a limited pilot sample size, it nonetheless provides novel insights into the use of peers in suicide prevention as one of the few peer-support post-hospitalisation programs to report a significant decrease in participants’ suicidal thoughts or behaviours [23,53]. The significant improvements in the hope scores after participating in PAUSE are consistent with other peer program evaluations [23,54]. Increasing hope has been posited as a key mechanism for peer program effectiveness, as peer workers can function as a type of “role model” to demonstrate that recovering from mental illness and suicidality is possible, which can increase participants’ hope and belief in their own ability to achieve recovery goals [24,26]. 

PAUSE peer support was experienced as an acceptable follow-up model for those who received support following a suicide-related hospitalisation. While the numbers in the current study are small, all survey and interview participants reported that PAUSE had helped them and that they would want someone important to them to be supported by PAUSE if experiencing suicidality. Additionally, there were no differences in the likelihood of engagement based on gender, mental illness history, cultural and linguistic diversity, or gender identity, suggesting acceptability of the PAUSE program within many populations for which effective and appropriate suicide prevention models are limited [55,56]. Continued analyses are required to examine these pilot findings. The absence of gender engagement differences in this pilot evaluation is promising, as men die by suicide at over twice the rate of women, but are less likely to engage with mental health treatment [2,57,58]. Effective interventions to address this disparity are critical for reducing suicide rates [59,60]. Expected judgement, reluctance to disclose information, and internalised stigmas have been reported as barriers to men help-seeking after suicide attempts [57]. Similar to other peer worker evaluations, the PAUSE interview participants reported non-judgemental attitudes as a key benefit to their experience with PAUSE [51], suggesting that peer support may be a potential acceptable model for addressing some of the barriers to engagement for men experiencing suicidality. Aboriginal and Torres Strait Islander people engaged with PAUSE at a lower, albeit not statistically different, rate (*p* = 0.066). As the number of First Nations people referred to the program was small, continued data collection is required to monitor and understand this trend. Further strategies may be needed to build trust and awareness before people are referred to PAUSE, such as engaging Aboriginal and Torres Strait Islander peer workers or partnering with Aboriginal community health organisations and community Elder groups. 

The peer skills and techniques that participants identified as most beneficial were the actions pertaining to feeling valued or connected with their PAUSE worker. Interestingly, though interviewees identified peer workers understanding them due to their similar experiences as one of the most effective aspects of PAUSE, workers discussing their illness and recovery strategies was identified as the least effective. Potentially, these seeming contradictions reflect that knowing that a peer worker has had their own experiences of mental illness and suicidality is critical to the greater trust conferred to peer workers, but only limited details of such experiences may be needed to gain this trust. Additionally, interview participants identified that as a result of their lived experience, peer workers treated them differently than other service providers, yet few reported that hearing about a peer worker’s personal experience was a factor in PAUSE’s impact. These differences may alternatively reflect that there are other places, such as support groups, where people can share with others impacted by suicidality and hear other recovery journeys, whereas peer worker support may be unique in that people can similarly connect with others with shared histories, but receive personal and individualised case support. This may be a crucial difference for people who have experienced a recent suicide attempt and many require immediate crisis assistance, but at that time may be unable to engage in support groups where distress may arise from other individuals’ stories. 

A surprising result from the thematic analysis was the broad range of support provided to participants, particularly the number who described peer workers assisting them with social housing waitlist applications as a significant part of the support they received. As the relationships between suicide risk and contextual stressors such as housing stress is well established, it may not be surprising that many PAUSE participants experiencing a suicidal crisis were also facing homelessness or housing instability [61]. This highlights the need for follow-up models that provide broad holistic assistance commensurate with the complex socio-environmental aetiology of suicidality [16,62], to which peer support may be a promising model offering understanding, advice, and practical assistance in dealing with serious hardships, such as stigmas, discrimination, and living in poverty, which have been identified as central, and often unique, benefits of peer worker support [28,29].

Though the original PAUSE pilot protocol planned to support people until they were connected to health and/or social services, this model was seldom requested by participants. While most were connected to other services through PAUSE, many people still required peer worker support, as it was framed as serving different purposes and addressing different needs. Some participants were connected to service providers, but still needed peer worker to coordinate and manage them—especially while they were acutely suicidal; others required social connection from peer workers to alleviate their isolation—which was described as a contributor to their suicidal crisis—in addition to social or mental health service providers. The perceived resourcefulness or willingness to help with what was needed was broadly observed as a key benefit of peer support for suicidality.

While many of these results may be generalisable to peer work models broadly, it is pertinent to note, however, that a number of the key factors identified may be specific to this context where peer worker services were delivered from a wholly lived-experience governed organisation. For example, participants identified that the flexible and responsive nature of PAUSE support—which often exceeded the initial program timeframes—was critical to its effect. This may not be possible with peer models delivered within mainstream services, which may adhere to delivery timeframes more rigorously. Additionally, ongoing social connectedness with PAUSE workers was identified as a key factor for the program’s effectiveness at reducing suicidality. Interviewees reported that they saw themselves as part of the Brook RED community and were able to stay connected through other community activities at Brook RED centres. The role that peer support may play in increasing social connection and reducing suicidality may not be replicated within different organisational structures where ongoing connection is not possible.

### Limitations 

The small number of participants that completed both pre- and post-questionnaires limited the generalisability of the findings to all PAUSE participants. With the large number of people who were lost to follow-up, those who did not experience improvements may have been likely to disengage with the evaluation. Questionnaire completion was also lower in the initial months of the pilot as PAUSE workers simultaneously juggled establishing the program and incorporating the evaluation activities. Furthermore, many people did not have a formal “service exit,” as they gradually and organically reduced their engagement with their peer worker once they were able to cope and manage independently without support. While the assistance required was often time-intensive immediately following hospital presentations, over time, participants often required—or indeed wanted—less support, took longer to respond to workers, and eventually stopped responding. PAUSE workers found it difficult to predict when a participant’s involvement with PAUSE would end and, in turn, when to complete questionnaires, or they found it difficult and uncomfortable to “hassle” people who were coping well and did not wish to be reminded of their previous attempt. 

Additionally, without a comparison control group, it is possible that the positive outcomes were due to potential natural decreases in suicidality over time rather than intervention effects. While the interview participants reported that suicide attempts and emergency presentations had decreased, corroboration through hospital data linkages has yet to occur. Further research is needed using study designs producing stronger evidence, including comparisons to “treatment as usual” and examinations of longer post-program effects.

## 5. Conclusions

Overall, the evaluation data suggested that the pilot delivery of the PAUSE peer support program was an acceptable and effective model for providing care in the critical period following suicide-related hospital presentations. The findings reflected that the participants experienced decreased suicidal ideation and increased hope for the future after working with a peer worker. The engagement rates and participant feedback suggested acceptability of the program for people experiencing current suicidality, including populations for which effective suicide prevention models are limited. The results from the experience questionnaires and interviews reflected that the participants attributed the peer worker program towards their reduced suicidality. The thematic analysis revealed that the reasons why the program had assisted them were holistic and responsive support and ongoing social connectedness with peer workers who understood their experiences and treated them like people rather than clients. The interview findings also reflected that the participants believed that PAUSE, and peer worker programs more broadly, helped to address unmet needs in supporting people experiencing suicidal crises.

## Figures and Tables

**Figure 1 ijerph-20-03763-f001:**
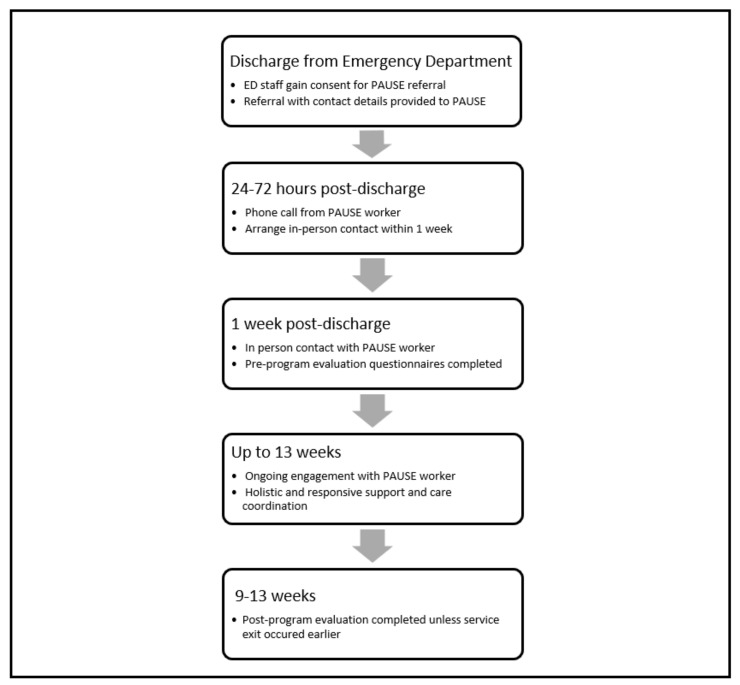
PAUSE pilot program workflow.

**Table 1 ijerph-20-03763-t001:** Participant demographic information and reason for referral by referral and engagement with PAUSE.

	*Referred* *N*	*Engaged* *N*	*X* ^2^	*df*	*p*
Gender			1.528	1	0.466
Men	127	54			
Women	186	88			
Indigenous status			3.376	1	0.066
First Nations	16	4			
Non-Indigenous	232	113			
CALD background			1.615	1	0.204
CALD identification	37	22			
Other	265	120			
Referral Reason			1.887	2	0.389
Mental illness history	58	23			
Situational crisis	81	41			
Both	58	29			

Missing data excluded from table; First Nations = identifies as an Aboriginal and/or Torres Strait Islander person; CALD = identifies as being from culturally and linguistically diverse populations.

**Table 2 ijerph-20-03763-t002:** Pre-intervention/baseline outcome measure mean scores.

Outcome Measure	*N*	*M*	*SD*
GHQ-28-SS	53	9.15	3.02
AHS	53	31.1	14.16
K10	36	32.99	7.89

GHQ-28-SS = general health questionnaire-28 suicide subscale; AHS = adult hope scale; K10 = Kessler psychological distress scale.

**Table 3 ijerph-20-03763-t003:** Pre-to post-PAUSE participant’s suicidal ideation, hope, and psychological distress.

Variable	*N*	Pre		Post		Wilcoxon Signed-Rank Test	
		*M*	*SD*	*M*	*SD*	*Z*	*p*
GHQ-SS	33	9.30	3.15	4.52	3.09	4.842	<0.001 *
AHS	32	32.49	13.2	39.19	11.04	2.789	0.005 *
K10	21	38.48	9.37	32.05	10.85	2.627	0.009 *

* < 0.05; GHQ-28-SS = general health questionnaire-28 suicide subscale; AHS = adult hope scale; K10 = Kessler psychological distress scale.

**Table 4 ijerph-20-03763-t004:** Peer skills or techniques experienced by participants.

Peer Action	Experienced by Participants	Action Effectiveness		
		Not at All	A Little/Fairly	Very Helpful
My PAUSE worker shared aspects of their own illness or recovery story with me.	82.4%	0%	42.3%	57.7%
My PAUSE worker told me about strategies that they have tried or used.	86.1%	7.4%	40.7	51.9%
My PAUSE worker understood my experiences because they have had similar experiences.	85.7%	0%	33.3%	66.7%
My PAUSE worker made me feel like the work I do toward my recovery is valid and valued.	100%	3.4%	13.8%	82.8%
I felt a sense of belonging with my PAUSE worker.	94.3%	0%	22.2%	77.8%
I felt a sense of connection with my PAUSE worker.	100%	0%	27.6%	72.4%
My PAUSE worker learned some things from me when we worked together.	91.4%	0%	38.4%	61.5%
My peer worker helped me to see that I have a lot of skills for recovery and life.	94.4%	0%	39.3%	60.7%
Knowing that my peer worker has overcome challenges made me feel like I could too.	91.7%	10.3%	24.1%	65.5%

## Data Availability

Restrictions apply to the availability of data. Data was obtained through Brook Red collection procedures. Data is not available due to participant privacy and ethical agreements.

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
