# Peer review of "Peer Intervention following Suicide-Related Emergency Department Presentation: Evaluation of the PAUSE Pilot Program"

_ijerph, 2023, doi:10.3390/ijerph20043763_

Round 1

Reviewer 1 Report

This is a generally well-written paper evaluating the effectiveness and acceptability of the PAUSE pilot program to prevent suicide-related crises and improve mental health among people with suicidal ideations and behaviors. To further enhance this manuscript, the reviewer had the following comments for the authors:

#1. Line 101-124: If feasible, it would be helpful to add a visual presentation to illustrate the workflow of the PAUSE program in conjunction with the text.

#2. Line 576: the reviewer could not retrieve citation #38, which seems to be a previous evaluation of the PAUSE pilot program suggested by the title. Please confirm the accuracy of the reference. If assessment of PAUSE was performed and published previously, please elaborate more on the additional information added by this submission.

#3. Consider adding a description of the study's statistical analyses in the Methods section.

#4. 3.1 Participants: the description of gender proportion between lines 184 and 186 appears to be confusing. Please confirm the proportion values. Summarizing the relevant sociodemographic and clinical characteristics as a table can be more intuitive.

#5. Discussion: the reviewer suggested the authors start the discussion section with a brief summary of this study and the highlights of key findings.

Minor: 

#6. Line 44-49: please consider polishing the first two sentences to improve their readability. 

#7. Line 199: the term “matched pairs” may not be straightforward to the audience, please consider further describing it.

#8. Line 209: consider using scientific notation when p value is very small to replace “p=.000”. 

#9. Line 246: please double-check this sentence and consider modifying it if applicable.

Author Response

Reviewer 1 Comments and Suggestions for Authors 

This is a generally well-written paper evaluating the effectiveness and acceptability of the PAUSE pilot program to prevent suicide-related crises and improve mental health among people with suicidal ideations and behaviors. To further enhance this manuscript, the reviewer had the following comments for the authors: 

Comment #1. Line 101-124: If feasible, it would be helpful to add a visual presentation to illustrate the workflow of the PAUSE program in conjunction with the text.  

Author response: A visual figure has been included to illustrate the processes and timeframes of the PAUSE pilot program activities. 

Comment #2. Line 576: the reviewer could not retrieve citation #38, which seems to be a previous evaluation of the PAUSE pilot program suggested by the title. Please confirm the accuracy of the reference. If assessment of PAUSE was performed and published previously, please elaborate more on the additional information added by this submission.  

Author response: Thank you for drawing our attention to the missing link in the citation which has now been included. The document is not previously published in a journal. As we had some delays submitting this manuscript to an open-access journal there was no available information about the program experience and outcomes for the local community. The document referenced here was created and made available to community stakeholders on the Brook RED website. It was identified as an important component of the lived experience ethos of community accountability and transparency for people with their own lived experience of mental health challenges or suicidality in the local community who might need to access the service to be able to learn about the program and other peoples' experiences. The document on the website sought to have more visual information and non-technical language with broader information about the program for people in the area. As there would be some overlap with this document and the manuscript, reference to the community document could be removed if that would be preferable. It may also be possible to remove the document from the website as the manuscript will be open-access available to the community. 

Comment #3. Consider adding a description of the study's statistical analyses in the Methods section.  

Author response: Thank you for this comment. This has now been included in the Materials and Methods section. 

Comment #4. 3.1 Participants: the description of gender proportion between lines 184 and 186 appears to be confusing. Please confirm the proportion values. Summarizing the relevant sociodemographic and clinical characteristics as a table can be more intuitive.  

Author response: Thank you for this suggestion. A table has been including to summarize demographic and clinical information at the beginning of the Results section 

Comment #5. Discussion: the reviewer suggested the authors start the discussion section with a brief summary of this study and the highlights of key findings.  

Author response: A summary of the aim, methodology and key findings has been included at the beginning of the Discussion section. Thank you for this suggested improvement. 

Minor:  

Comment #6. Line 44-49: please consider polishing the first two sentences to improve their readability.  

Author response: These sentences have been edited to read ‘An issue identified by people who have experienced suicidality is that the treatment they received often focused on mental illness diagnoses or risk assessments to the exclusion of precipitants or triggers contributing to their suicidality or the contextual stressors they experienced – despite the well-documented association between social and environmental contextual factors, such as poverty, unemployment, interpersonal violence and increased suicide risk’ 

Comment #7. Line 199: the term “matched pairs” may not be straightforward to the audience, please consider further describing it. 

Author response: The wording has been clarified to explain that it pertains to cases where both pre- and post- program evaluation questionnaires are available. 

Comment #8. Line 209: consider using scientific notation when p value is very small to replace “p=.000”.  

Author response: Thank you for drawing our attention to this. To avoid very small numbers or values = 0, we have clarified this to report <.001 

Comment #9. Line 246: please double-check this sentence and consider modifying it if applicable. 

Author response: Thank you. This sentence has been clarified to ‘Ten (six men and four women) previous PAUSE participants were interviewed.’  

Reviewer 2 Report

The paper investigates the efficacy of the PAUSE program in reducing suicidal ideation among patients presenting with suicidal ideation in the emergency departments (ED). The findings from the mixed methods study, which includes 142 patients report promising findings where participants report a significant decrease in suicidal ideation scores and an improvement in hope scores. Additionally, the authors implemented a thematic content analysis to identify the key factors contributing to the efficacy of the program. The paper makes an important contribution and investigates a crucial topic; however, I have a few concerns that I would like the authors to address.

General Comments

·         There are several typos and wording issues. For example, I am not sure what the authors mean in lines 32 and 33 where they say “the previous year prior”.

·         Line 33 replace “is increased” with “increases”

·         Line 48-49. Please rewrite this sentence. There are a lot of sentences throughout the manuscript which is choppy and do not flow well. Although the manuscript has the core content, the writing lack coherence. Please have a technical writer review the manuscript before submitting the revision.

·         Table 1. Is there a particular reason that authors performed the Wilcoxon-Signed Rank test rather than performing a paired t-test? Also, can the authors report the 95% CI (assuming alpha was 0.05) for the mean difference between groups?

·         I am not sure what the authors mean by this statement “36 questionnaires were collected in total” Line 226

·         Line 246-247 – It should be “were interviewed.”

·         Line 247 authors mention PAUSEA? Typo?

·         Overall, the methods and design are robust. Also, the reported findings are insightful and important, but manuscript quality in terms of writing needs to be significantly improved. 

Author Response

REVIEWER TWO: The paper investigates the efficacy of the PAUSE program in reducing suicidal ideation among patients presenting with suicidal ideation in the emergency departments (ED). The findings from the mixed methods study, which includes 142 patients report promising findings where participants report a significant decrease in suicidal ideation scores and an improvement in hope scores. Additionally, the authors implemented a thematic content analysis to identify the key factors contributing to the efficacy of the program. The paper makes an important contribution and investigates a crucial topic; however, I have a few concerns that I would like the authors to address. 

General Comments 

Comment There are several typos and wording issues. For example, I am not sure what the authors mean in lines 32 and 33 where they say “the previous year prior”. 

Author response: Thank you for identifying this error. The word ‘prior’ has been removed. 

Comment Line 33 replace “is increased” with “increases” 

Author response: AMENDED 

Comment Line 48-49. Please rewrite this sentence. There are a lot of sentences throughout the manuscript which is choppy and do not flow well. Although the manuscript has the core content, the writing lack coherence. Please have a technical writer review the manuscript before submitting the revision. 

Author response: These lines have been clarifying to instead say ‘Indeed, previous research has reported that people experiencing these adverse life events and stressors were more likely to die in the weeks following discharge from psychiatric admissions than those who were discharged who were not also experiencing such stressors’ 

Thank you for this suggestion. The manuscript has been reviewed independently for writing clarity with changes noted throughout the manuscript. 

Comment Table 1. Is there a particular reason that authors performed the Wilcoxon-Signed Rank test rather than performing a paired t-test? Also, can the authors report the 95% CI (assuming alpha was 0.05) for the mean difference between groups? 

Author response: The non-parametric Wilcoxon-Signed Rank test was used over the paired t-test as data within the small pilot sample was not normally distributed. Normality was not assumed as the participants were all recruited following so skewness towards higher distress and suicidality was anticipated. This has been further clarified in the methods section. 

Comment  I am not sure what the authors mean by this statement “36 questionnaires were collected in total” Line 226 

Author response: This line has been clarified to say ‘PAUSE Experience questionnaires were collected from a total of 36 PAUSE participants.’ 

Comment Line 246-247 – It should be “were interviewed.” 

Author response: AMENDED. Thank you for picking this up this error. 

Comment Line 247 authors mention PAUSEA? Typo? 

Author response: AMENDED. Thank you for picking this up this error. 

Reviewer 3 Report

This is a very interesting study of evaluation of the PAUSE pilot program. However, there are few things that should be clarified.

1.      In the Materials and Method section, it should be added participants or study population, and explained how participants were selected. We can see at the beginning of the Results section how participants were recruited, but that section should be expanded in the Method. It was reported that 315 people were referred from the Emergency Department of the Logan General Hospital following suicide-related presentations, but how that 315 people were selected. Were they consecutive patients from Emergency department or were they selected on some basis? Also, it would be good to know which population is hospitalized in Logan General Hospital, are they from one city or region, or the whole country?

2.      In the section Evaluation measures (line 130-135) it should be mentioned what does it mean? Did every participant received GHQ 28-SS, AHS and K10?

3.      Also, it should be explained when and where participants fulfilled post-intervention questionnaires.

4.      In the method section should be mention which statistical tests were used in the data analysis.

5.      In the result section should be presented one table with participantsˋ characteristics. It is difficult to follow text (lines 183-196) without data presented. Also, from the discussion section we could see that Aboriginal and Torres Strait Islander people were less likely to engage with PAUSE, but it is not presented like that in the results section. Please, provide detailed demographic data for all participants.

6.      Also, should be explained what does it mean, mean baseline AHS and K10 scores? Are those scores from the questionnaires received pre-intervention and how many participants participated because we can see from the table one that baseline scores are not the same as the pre-intervention scores presented in the table?

7.      It would be useful to know when, where and how semi-structured interviews were conducted and how many questions included.

8.      Line 246 ”Ten (six men and four women and six men)”, six men were mentioned two times.

9.      References were not uniformly cited, please cite them in the same manner.

Author Response

REVIEWER THREE 

This is a very interesting study of evaluation of the PAUSE pilot program. However, there are few things that should be clarified. 

Comment 1.      In the Materials and Method section, it should be added participants or study population, and explained how participants were selected. We can see at the beginning of the Results section how participants were recruited, but that section should be expanded in the Method. It was reported that 315 people were referred from the Emergency Department of the Logan General Hospital following suicide-related presentations, but how that 315 people were selected. Were they consecutive patients from Emergency department or were they selected on some basis? Also, it would be good to know which population is hospitalized in Logan General Hospital, are they from one city or region, or the whole country? 

Author response: Information about Logan hospital has been included in the Materials and Methods section. Participant selection information has also been included as follows ‘All people who presented to the Logan General Hospital – Emergency Department for suicidal ideation, self-injury, or suicide attempt were provided information about the PAUSE program and were invited to engage with a peer-support program to provide follow-up after presentation. Those who consented were referred to PAUSE by Emergency Department staff providing participant contact details to PAUSE staff.’ 

Comment 2.      In the section Evaluation measures (line 130-135) it should be mentioned what does it mean? Did every participant receive GHQ 28-SS, AHS and K10? 

Author response: The inclusion of all three measures in the evaluation questionnaire at both pre- and post- intervention time points has been clarified in the methods section. 

Comment 3.      Also, it should be explained when and where participants fulfilled post-intervention questionnaires. 

Author response: This has been clarified by adding ‘Participants were asked to complete the post- program evaluation questionnaire between 9-13 weeks after first engaging with a PAUSE worker, unless the participant exited the service prior to this point.’ 

Comment 4.      In the method section it should be mention which statistical tests were used in the data analysis. 

Author response: A statistical analyses section has been included at the end of the materials and methods section 

Comment 5.      In the result section should be presented one table with participants characteristics. It is difficult to follow text (lines 183-196) without data presented. Also, from the discussion section we could see that Aboriginal and Torres Strait Islander people were less likely to engage with PAUSE, but it is not presented like that in the results section. Please, provide detailed demographic data for all participants. 

Author response: Further explanation has been included in both the results and discussion. A table has been included in the results section to provide demographic data for all participants. 

Comment 6.      Also, should be explained what does it mean, mean baseline AHS and K10 scores? Are those scores from the questionnaires received pre-intervention and how many participants participated because we can see from the table one that baseline scores are not the same as the pre-intervention scores presented in the table? 

Author response: An additional table has been included to clarify the pre-intervention scores. Additional wording has been added to explain that the difference on the scores for all pre-intervention questionnaires received and for those for whom both pre- and post-questionnaires were collected. 

Comment 7.      It would be useful to know when, where and how semi-structured interviews were conducted and how many questions included. 

Author response: Information about the interview questions has been included, the key questions were: What support or assistant did you PAUSE worker provide? What was your experiences of being supported by a peer worker with their own lived experience? What could be done to improve the program going forward? Would you want someone you cared about to access PAUSE if there experienced a suicidality crisis?  

Text explaining interview procedures has also been included as follows, “Participants were offered to have the interview conducted over the phone, in-person at a location of their choosing or at the researcher’s offices. One participant chose to be interviewed at the researcher’s offices with the remaining all electing to be interviewed over the phone.” 

Comment 8.      Line 246 ”Ten (six men and four women and six men)”, six men were mentioned two times. 

Author response: AMENDED. Thank you for picking this up this error. 

Comment 9.      References were not uniformly cited, please cite them in the same manner. 

Author response: Thank you. The references have been reformatted and double-checked. 

Round 2

Reviewer 1 Report

Dear authors,

Thank you for taking my suggestions into consideration. I do not have any further questions. It's my pleasure to review this great work!

Reviewer 3 Report

The manuscript has been significantly improved.